# Effect of Chitin Nanocrystals on Crystallization and Properties of Poly(lactic acid)-Based Nanocomposites

**DOI:** 10.3390/polym12030726

**Published:** 2020-03-24

**Authors:** Shikha Singh, Mitul Patel, Daniel Schwendemann, Marta Zaccone, Shiyu Geng, Maria Lluisa Maspoch, Kristiina Oksman

**Affiliations:** 1Division of Materials Science, Luleå University of Technology, SE-97 187 Luleå, Sweden; shikha.singh@ltu.se (S.S.); mitul.patel@ltu.se (M.P.); daniel.schwendemann@hsr.ch (D.S.); shiyu.geng@ltu.se (S.G.); 2Centre Català del Plàstic (CCP), Universitat Politècnica de Catalunya Barcelona Tech (EEBE-UPC), C/Colom 114, Terrassa 08222, Spain; maria.lluisa.maspoch@upc.edu; 3IWK Institut für Werkstofftechnik und Kunststoffverarbeitung, CH-8640 Rapperswil, Switzerland; 4Proplast, Via Roberto di Ferro 86, 15122 Alessandria, Italy; marta.zaccone@proplast.it; 5Mechanical & Industrial Engineering, University of Toronto, Toronto, ON M5S 3BS, Canada

**Keywords:** poly(lactic acid), chitin nanocrystals, nanocomposites, liquid-assisted extrusion, crystallinity, barrier properties, hydrolytic degradation

## Abstract

The crystalline phase of poly(lactic acid) (PLA) has crucial effects on its own properties and nanocomposites. In this study, the isothermal crystallization of PLA, triethyl citrate-plasticized PLA (PLA–TEC), and its nanocomposite with chitin nanocrystals (PLA–TEC–ChNC) at different temperatures and times was investigated, and the resulting properties of the materials were characterized. Both PLA and PLA–TEC showed extremely low crystallinity at isothermal temperatures of 135, 130, 125 °C and times of 5 or 15 min. In contrast, the addition of 1 wt % of ChNCs significantly improved the crystallinity of PLA under the same conditions owing to the nucleation effect of the ChNCs. The samples were also crystallized at 110 °C to reach their maximal crystallinity, and PLA–TEC–ChNC achieved 48% crystallinity within 5 min, while PLA and PLA–TEC required 40 min to reach a similar level. Moreover, X-ray diffraction analysis showed that the addition of ChNCs resulted in smaller crystallite sizes, which further influenced the barrier properties and hydrolytic degradation of the PLA. The nanocomposites had considerably lower barrier properties and underwent faster degradation compared to PLA–TEC110. These results confirm that the addition of ChNCs in PLA leads to promising properties for packaging applications.

## 1. Introduction

Poly(lactic acid) (PLA)-based nanocomposites have been widely researched owing to their potential applications. The mechanical properties of PLA are comparable to those of polystyrene (PS) and polyethylene terephthalate (PET), that are commercial polymers used for packaging applications. PLA has good optical properties. However, PLA has slow crystallization rate, moderate gas permeability, and low elongation at break that limits its use in the packaging industry [1,2].

Crystallinity plays a vital role in the improvement of mechanical, thermal, optical, and barrier properties of polymers. Usually, nucleating agents are added to the polymers to increase their crystallinity [3]. Chitin whiskers (ChNWs), also known as chitin nanocrystals (ChNCs), were first isolated from a crab shell by Marchessault et al. [4] in 1959 via acid hydrolysis. ChNCs have attracted significant attention as nucleating agents because of their natural origin, low toxicity, low density, large surface area, and biodegradability [5]. ChNCs exhibit a rod-like shape, have a length of 50–300 nm, a diameter of 10–30 nm, an aspect ratio of approximately 15, and a modulus of 150 GPa [6]. Furthermore, ChNCs have acid amide functional groups on the surface that provide better scope of interactions with the polymers [7]. Additionally, they possess antibacterial [8] and antifungal properties [9] that further expand their usage for packaging applications.

Polymer nanocomposites containing ChNCs were first prepared by Paillet and Dufresne, in 2001 [6]. Initially, ChNCs were incorporated into poly(styrene-co-butyl acrylate) nanocomposites [6]. Subsequently, ChNCs were successfully added into several other polymers [10,11,12]. Morin and Dufresne prepared a nanocomposite adding ChNC in poly(caprolactone) (PCL) and observed that ChNC increased the relaxed modulus of PCL/ChNC composites from 0.6 MPa to 6.8 MPa [10]. Lu et al. [11] added ChNCs in a soy protein isolate (SPI) matrix and reported that ChNC (20 wt %) increased the mechanical properties of the SPI/ChNC nanocomposites. The tensile strength increased from 4 MPa to 8 MPa and Young’s modulus increased from 26 MPa to 158 MPa. Wang et al. [13] reported that on adding 0.5 wt % of surface-modified ChNCs into poly(3-hydroxybutyrate-co-3-hydroxyvalerate) (PHBV), the ChNCs improved both the Young’s modulus and strength of the PHBV composites by 44% and 67%, respectively.

The development of PLA nanocomposites with ChNCs [9,14,15,16] has attracted considerable interest. Generally, PLA nanocomposites have been prepared by solvent casting technique because solvent casting is easy and the dispersion of ChNC into dissolved polymer is convenient as well, but it is difficult to upscale and also organic solvents are used if non-water-soluble polymer is used as matrix. The main challenge in the preparation of PLA nanocomposites with ChNCs is to develop up-scalable processing methods where ChNCs are homogenously dispersed into PLA [14]. Poor compatibility between the hydrophobic PLA and hydrophilic ChNC resulted to the poor dispersion which eventually forms a weak polymer-nanoreinforcement interactions and thus, formation of agglomeration of the ChNCs are found. Oksman et al. [17], used liquid-assisted extrusion to overcome this problem of dispersion of hydrophilic nano-reinforcements into a hydrophobic polymer matrix and successfully prepared well-dispersed cellulose nanocrystal (CNC) PLA nanocomposites. Later, they used the same technique to prepare PLA nanocomposites using ChNCs and CNCs [18]. The compositions of PLA, triethyl citrate (TEC), and ChNC in the nanocomposites were in the proportions of 79:20:1(wt %) [18]. Microscopy revealed a number of small aggregates in the nanocomposites. However, it was observed that the ChNC and CNC nanocomposite films prepared with slow and fast cooling rates affected the final properties of the PLA, and the addition of ChNC resulted in better properties than the addition of CNC [18]. In another study, the process was upscaled with film blowing and PLA/ChNC nanocomposite films were successfully produced using 6 wt % of TEC and 1 wt % of ChNC [19]. They reported that ChNCs acted as a multifunctional additive that increased the viscosity, melt strength, thermal stability, and the crystallinity. In addition, an increase in tear strength and puncture strength (175% and 300%, respectively) in PLA/ChNC nanocomposites was reported [19].

We previously conducted a detailed investigation on the effect of ChNCs on the crystallization behavior (crystallization rate, kinetics, crystal types) of TEC-plasticized PLA [20]. It was observed that a very low amount of ChNC (1 wt %) increased the crystallization rate by acting as an excellent nucleating agent, and therefore reduced the overall crystallization time of the plasticized PLA.

The aim of the present study is to further explore the knowledge of crystallization and investigate how ChNC affects the crystallinity and the thermal, optical, barrier, and hydrolytic degradation properties of nanocomposites. Isothermal crystallization was carried out at different temperatures (135, 130, 125, and 110 °C) and holding times (5, 15, and 40 min) using compression molding. The morphology and dimensions of the ChNCs were analyzed by atomic force microscopy (AFM). Scanning electron microscopy (SEM) was used to analyze the dispersion and morphology of the materials. Fourier transform infrared (FTIR) spectroscopy was used to investigate the interaction between ChNC and PLA during the isothermal crystallization of materials. The thermal properties and degree of crystallinity were investigated using differential scanning calorimetry (DSC). X-ray diffraction (XRD) was used to investigate the crystal structure and polarized optical microscopy (POM) was used to determine the spherulite size. The barrier properties of the materials were tested with water vapor transmission rate (WVTR) and oxygen permeability (OP) tests. The effect of crystallinity induced by ChNCs on the hydrolytic degradation was further examined. Finally, the influence of hydrolytic degradation on the thermal stability of the materials was investigated by thermogravimetric analysis (TGA).

## 2. Materials and Methods

### 2.1. Materials

Polylactic acid (PLA) in pellet form (Ingeo 4043D) from NatureWorks, (Minnetonka, MN, USA) was used as the matrix. Chitin powder from shrimp shell was purchased from Sigma-Aldrich (grade C7170 (Stockholm, Sweden) and used as a starting material for the isolation of ChNCs. HCl (ACS reagent, 37%) for acid hydrolysis was purchased from Merck (Darmstadt, Germany). TEC (*M*_w_: 276.3 g/mol) in liquid form was purchased from VWR (Stockholm, Sweden), and ethanol (99.5%) was purchased from Solveco (Stockholm, Sweden).

### 2.2. Preparation of ChNCs

ChNCs were isolated via hydrochloride acid hydrolysis treatment according to the procedure described by Herrera et al. [21]. Briefly, the chitin powder was hydrolyzed using 3 M HCl at 90 ± 5 °C under vigorous stirring for 90 min. The ratio of acid to chitin solids was 30 mL per gram of chitin. After acid hydrolysis, the suspension was diluted with distilled water and subjected to centrifugation at 8000 rpm for 10 min. The supernatant after centrifugation was decanted and the precipitate was diluted again with distilled water. This centrifugation process was repeated three times. Afterwards, the suspension was transferred to dialyze for 5 days. For the disintegration of the remaining large particles, the suspension was subjected to ultrasonication treatment for 20 min. The final suspension was then evaporated to obtain a ChNC gel with a solid content of 18 wt % and was subsequently stored at 4 °C for later use.

### 2.3. Preparation of Nanocomposite Pellets via Liquid-Assisted Extrusion

PLA nanocomposites were prepared by melt compounding via liquid-assisted extrusion as reported by Oksman and co-workers [17,19,21]. In this method, ChNCs were fed in liquid form and the suspension was prepared as follows: ChNC gel in water (18 wt %) was pre-dispersed in a water/ethanol solvent mixture with a weight ratio of 1:5 for 2 h via magnetic stirring and then mixed with TEC (2.61 wt % solid content was added to achieve 10 wt % of TEC). To feed the suspension into the extrusion a peristaltic pump PD 5001 Heidolph (Schwalbach, Germany) was used. The specific feeding rates of the PLA and the suspension as well as the final composition of materials are displayed in Table 1.

A co-rotating twin-screw extruder ZSK-18 MEGALab, Coperion W&P (Stuttgart, Germany) length to diameter screw ratio (L/D) 40 and screw diameter 18 mm, equipped with K-tron gravimetric feeder (Niederlenz, Switzerland) was used for the production of nanocomposites. A schematic representation of the nanocomposite pellet preparation process is shown in Figure 1a. Finally, pellets of PLA–TEC–ChNC with 1 wt % of ChNCs and neat PLA and PLA–TEC as references were prepared for later processing.

### 2.4. Preparation of Isothermal-Crystallized Films

PLA, PLA–TEC, PLA–TEC–ChNC films were prepared by compression molding using a laboratory press LPC-300 Fontijne Grotnes (Vlaardingen, Netherlands). For this, 4 g of each material was placed between two metallic sheets to mold the films. First, the material was preheated at 190 °C and was then compression-molded using a pressure of 8.2 MPa for 1 min. Thereafter, the film was cooled to the isothermal crystallization temperature (135 °C, 130 °C, 125 °C, or 110 °C) and kept for 5, 15, or 40 min, and was subsequently cooled to room temperature using a water cooling system equipped with the laboratory press (Figure 1b). The prepared samples were coded according to their corresponding isothermal crystallization temperature (*IC*_TEMP_) and time (*IC*_TIME_), as shown in Table 2.

### 2.5. Characterizations

Atomic force microscopy (AFM) was used to characterize the morphology and dimensions of the ChNCs using a Veeco Multimode Nanoscope (Santa Barbara, CA, USA) in the tapping mode. The length and diameter of the ChNCs were analyzed using the Gwyddion software version 2.55 (Czech Metrology Institute, Brno, Czech) [22].

Differential scanning calorimetry (DSC) Mettler Toledo 822e (Schwerzenbach, Switzerland) was used to study the crystallinity of the isothermally crystallized films. Additionally, the crystallinity of hydrolytic degraded films was investigated. Samples were placed in an aluminum crucible, and then analyzed under a nitrogen atmosphere from −20 to 200 °C at a heating rate of 10 °C/min. The crystallinity of the samples was calculated using following Equation [23]:(1)$$Crystallinity\left(\%\right)=\frac{\Delta H_{m}-\Delta H_{cc}}{93.1}\times \frac{100}{w}$$
where ∆*H*_m_ is the melting enthalpy, ∆*H*_cc_ is the cold crystallization enthalpy, the constant 93.1 (unit: J/g) corresponds to the Δ*H*_m_ for 100% crystalline PLA [23], and *w* is the weight fraction of PLA in the samples [23].

The thermal stability of the isothermally crystallized films was investigated using thermo-gravimetric analysis (TGA), TA Instrument Q500 (New Castle, DE, USA) under nitrogen atmosphere. Approximately 9 mg of material was subjected for testing at a heating rate of 10° C/min in a temperature range of 0–600 °C.

X-ray diffraction using a PANalytical Empyrean diffractometer (Almo, Malvern, UK)) was performed to investigate the crystallite size of the PLA. The measurements were performed with Cu–Kα radiation (λ = 1.5405 Å). An acceleration voltage of 45 kV and current of 40 mA were used over a range of 5°–45° with a step size of 0.026°. The crystallite size was investigated using the Scherrer Equation:(2)$$Crystallite\;size=\frac{k\lambda }{\beta \mathrm{Cos}\theta }$$
where *k* is the dimensional shape factor, and is 0.9 [24], λ is the wavelength, β is the full-width at half maximum for different peaks, and θ is the Bragg angle.

The light transmittance of the materials was measured using a UV–Vis spectrophotometer (GENESYS, 10 UV, Thermo-Scientific, Dreieich, Germany) at a constant wavelength of 550 nm, and three specimens from each sample were tested to calculate the average values.

Polarized microscopy (POM) Nikon Eclipse LV 100 Pol (Kanagawa, Japan), was used to study the spherulite morphology and size developed during isothermal crystallization.

Fractured surfaces of the PLA110, PLA–TEC110, and PLA–TEC–ChNC110 were studied by scanning electron microscopy (SEM) JEOL, JSM-IT300 (Tokyo, Japan). Prior to the study, the surfaces of the samples were sputter-coated (Leica EM ACE220, Wetzlar, Germany) with platinum to avoid the charging effect. The acceleration voltage was 15 kV, and secondary electron images were collected.

Fourier infrared spectroscopy with the attenuated total reflectance mode (ATR-FTIR) VERTEX 80, Bruker, (Ettlingen, Germany) was carried out to investigate the interaction between PLA, TEC, and ChNCs, induced during the isothermal crystallization of PLA110, PLA–TEC110, and PLA–TEC–ChNC110. The spectra were recorded in the wavenumber range of 400–4000 cm^−1^.

Water vapor transmission rate (WVTR) was measured using a modified method according to ASTM E96. Films were cut into circular discs with diameters of 0.04 m. The test samples were placed on a cup filled with silica gel and then placed in a chamber with controlled temperatures of 23 °C and 50% relative humidity (RH). The cups were weighed after specific time intervals and WVTR (g/m^2^ day) was determined using the following equation [25]:(3)$$\mathrm{WVTR}=\frac{G/t}{A}$$
where *G*/*t* is the slope of the curve with increased weight gain (g) as a function of time (h) and *A* is the exposed area (m^2^).

The oxygen permeability (OP) tests were performed on the PLA–TEC and PLA–TEC–ChNC films using a Multiperm 037 equipment (ExtraSolution, Pieve Fosciana, Italy), according to the ASTM F2622–08. The surface area of the formed square films was 2 cm^2^ and the thickness was approximately 120 µm; the films were previously conditioned for 12 h under a continuous flux of electronically controlled anhydrous nitrogen. This preliminary step is necessary to stabilize the specimens and to remove the oxygen already present inside the sample before the beginning of the test. The duration of this phase is strongly related to both the barrier properties and the thickness of the material under testing. The thicker the specimen, the longer will be the conditioning phase. Typically, the following empirical equation is used to calculate this duration:(4)$$Conditioning\;time\left[h\right]=\frac{thickness\;\left[\mu m\right]}{10}$$

At the end of the conditioning phase, the oxygen flux was determined for the surfaces of the specimens. A carrier collected and a sensor detected the amount of oxygen that permeated through the films. The test was performed at 23 °C and 50% RH. The oxygen flux of the film surfaces was maintained at 13.5 mL/min on an average. Two specimens were tested for each formulation. The reported data were referred to mediated values. The oxygen transmission rate (OTR) from the test corresponds to the oxygen permeability of the material and is calculated using following Equation:(5)$$\frac{OTR_{f}}{A_{ts}}=OTR$$
where *OTR* is the oxygen transmission rate [cc/m^2^ 24h], *A*_ts_ is the surface area of the test sample [m^2^], and *OTR*_f_ [cm^3^/day] is the final measured permeation concentration. This parameter is also calculated as:(6)$$OTR_{f}=OTR_{m}-OTR_{b}$$
where *OTR*_m_ is the measured oxygen transmission rate and *OTR*_b_ corresponds to the background oxygen transmission rate.

Hydrolytic degradation of the materials was performed according to the ASTM F163 on 30 mm × 30 mm × 0.1 mm films. The samples were dipped into distilled water kept inside in an oven set at 58 °C. Intermittently, the samples were taken out and gently wiped with tissue paper to remove the water droplets present on the surface, and the weights of the samples were recorded. The degradation process was monitored up to 18 days. Water uptake studies were also performed to investigate the diffusion kinetics of PLA110, PLA–TEC110, and PLA–TEC–ChNC110. Furthermore, the effect of hydrolytic degradation on thermal properties was studied.

## 3. Results

Inspired by our previous studies on liquid-assisted extrusion [18,19,21], we have successfully produced PLA/ChNCs nanocomposites by using a co-rotating twin-screw extruder. The liquid feeding of ChNCs along with TEC plasticizer lead to PLA–TEC–ChNCs nanocomposites with improved dispersion and distribution of ChNCs. Furthermore, it is very important to have a controlled and effective atmospheric venting, as well as a vacuum system to evacuate the vapor (550 g/hr) of liquid (water: ethanol) during the extrusion process. Therefore, a co-rotating twin-screw extruder is the best choice of equipment due to its excellent degassing properties.

### 3.1. Morphology of ChNCs and Visual Appearance of Neat Films

The size and shape of the nanoreinforcement play an important role in the nanocomposites. Therefore, the morphology of the ChNCs were examined using AFM and image displays rod-shaped ChNCs (Figure 2a). The length and diameter of the ChNCs were in the range of 273 nm and 11 nm, respectively, and the corresponding histograms are shown in Figure 2b–c. The very small diameter and high aspect ratio of the ChNCs ensure that they can present excellent functionalities, e.g., serving as reinforcements and improving barrier properties, when they are well-dispersed in the PLA–TEC matrix. Photographs and optical micrographs of the prepared neat PLA, PLA–TEC, and PLA–TEC–ChNC films (without isothermal crystallization) are presented in Figure 2d–f,d’–f’.

All films are clear and transparent, which is attributed to the fast cooling process during the compression molding resulting in very low crystallinity in all three samples (Table 3). This also confirms that the ChNCs were well-dispersed and distributed in the PLA–TEC matrix and no large agglomerates were visible under an optical microscope, as shown in Figure 2f’.

#### 3.1.1. Surface Morphology

The morphologies of the surface and cross-section of the fractured samples *IC*_TEMP_ at 110 °C were studied by SEM as shown in Figure 3. The fractured surface of the PLA110 film was relatively coarse, as evidenced by the surface view as well as the cross-sectional view. In contrast to PLA, the incorporation of TEC into PLA (i.e., PLA–TEC110) exhibited a homogenous behavior. In the surface view of PLA–TEC–ChNC110, a highly ordered pattern of spherulites was seen which may be due to the well-dispersed ChNCs resulting in the formation of homogenous spherulites. In the cross-sectional view of PLA–TEC–ChNC110, no agglomerates were found which is attributed to the homogenous dispersion and distribution of the ChNCs in the nanocomposite.

#### 3.1.2. Surface Interaction between PLA, TEC, and ChNC

Isothermal crystallization (especially at 110 °C) induced some interactions between the PLA, TEC, and ChNC which further investigated with ATR-FTIR and spectra of PLA110, PLA–TEC110, and PLA–TEC–ChNC110 are presented in Figure 4. All samples show a sharp peak at 1751 cm^−1^ that corresponds to the characteristic carbonyl peak of the PLA. The –C=O peak intensity of the nanocomposite was slightly reduced compared to PLA110 and PLA–TEC110. Peaks at 2997, 2954, 1453, 1386, 1358, 1266, 1128, 923, and 869 cm^−1^ are attributed to the asymmetric and symmetric –CH stretching, methyl bending, asymmetric and symmetric –CH bending, –C=O bending, –C–O– stretching, and –C–C– stretching (backbone) of PLA, respectively [26]. In PLA–TEC110, peaks observed at 3658 and 3506 cm^−1^ are attributed to –OH stretching [27]. The FTIR spectra of PLA–TEC110 showed certain molecular changes in the 2992–3509 cm^−1^ range that corresponds to –CH aliphatic stretching [27]. One new broad band appeared at 2925 cm^−1^. PLA–TEC–ChNC110 did not show significant differences but the overall intensity of the nanocomposites decreased and some overlapping bands were found in the fingerprint region. This may be due to certain molecular interactions that may have occurred between the components of the nanocomposites.

### 3.2. Thermal Properties, Crystallinity and Crystal Strucutre

The thermal properties of all samples with various *IC*_TEMPs_ and *IC*_TIMEs_, including the glass transition temperature (*T*_g_), cold crystallization temperature (*T*_cc_), melt temperature (*T*_m_), and crystallinity were determined from the first DSC heating scan, and the obtained data are summarized in Table 3. The crystallinity of the nanocomposites crystallized at 135, 130, or 125 °C for 15 min was much higher than that of neat PLA and PLA–TEC. At 110 °C, the nanocomposites achieved 47.5% crystallinity within 5 min of crystallization, while for PLA and PLA–TEC, 40 min was required to reach similar crystallinity. These indicate that the well-dispersed ChNCs can act as very effective nucleation agents for PLA crystallization. PLA had a constant *T*_g_ (approximately 59 °C) under all isothermal conditions, which is much higher than that of PLA–TEC (approximately 46 °C) owing to the presence of the plasticizer in PLA–TEC. PLA–TEC–ChNC showed a similar *T*_g_ as compared to PLA–TEC within 5 min of crystallization (at 135, 130, or 125 °C). However, the *T*_g_ of nanocomposites was relatively lower than that of PLA–TEC for 15 min of crystallization. This could be owing to the excellent nucleation ability of ChNCs resulting in higher crystallinity (increasing from 14.2% to 36.5%) under this condition. Interestingly, at the lowest temperature, i.e., 110 °C, PLA–TEC–ChNC exhibited the lowest *T*_g_ (37 °C). In addition, the *T*_cc_ of PLA–TEC was lower than that of neat PLA because of more flexible polymer chains, and the PLA–TEC–ChNC samples with higher crystallinity possessed lower *T*_cc_ owing to their more plasticized amorphous phase. PLA showed a *T*_m_ of approximately 169 °C at all isothermal conditions; on the other hand, there were no significant differences in the *T*_m_ of PLA–TEC and PLA–TEC–ChNC (approximately 164 °C), as observed under all isothermal conditions.

As similar crystallinity (45.5%–49.9%) was achieved at *IC*_TEMP_ of 110 °C for all three types of materials, i.e., PLA, PLA–TEC, and PLA–TEC–ChNC, the following characterizations focused on these three samples was to avoid further influence from different crystallinities. XRD analyses were carried out to investigate the crystal structure of the samples. Figure 5a shows the XRD patterns of PLA110, PLA–TEC110, and PLA–TEC–ChNC110. All materials exhibited peaks at 14.8°, 16.6°, 18.9°, and 22.2°, corresponding to the (200), (110), (203), and (015) planes of the PLA crystals, respectively. The crystallite sizes of the materials were calculated from the XRD patterns according to the Scherrer Equation (Equation (2)). The crystallite size of PLA–TEC was 14 nm, which was slightly larger than that of neat PLA (12 nm). PLA–TEC–ChNC showed the smallest crystallite size (8 nm), which is attributed to the nucleation effect of the ChNCs. The spherulite structures of these three samples were also studied by POM as shown in Figure 5c–e. Spherulites sizes of the PLA110, PLA–TEC110, and PLA–TEC–ChNC110 were measured to be 49, 56, and 23 µm, respectively, which are consistent with the XRD results.

### 3.3. Optical Properties

The optical transparency of the materials was investigated, and the light transmittance data are presented in Figure 5b and in Appendix A (see Appendix A). In general, samples with higher crystallinity exhibited lower transparency owing to the light scattering of the crystalline region in the materials. For the samples with 110 °C of *IC*_TEMP_ that showed similar crystallinity, the transparency was influenced by the crystallite size. As shown in Figure 4b, PLA110 exhibited a transmittance of 61% that was higher than that of PLA–TEC110 (58%), while PLA–TEC–ChNC110 showed the highest value of 69%. This is because of the well dispersed ChNC which resulted into the formation of homogenous crystallites. In addition, the photographs of samples with *IC*_TEMP_ of 110 °C shown in Figure 5c’–e’ illustrate that they were significantly opaquer compared to the films without isothermal crystallization (Figure 2d–f).

### 3.4. Barrier Properties

The barrier properties of the PLA–TEC110 and PLA–TEC–ChNC110 were investigated and the values of WVTR, OTR, and OP are summarized in Table 4. Compared to the WVTR and OTR values of amorphous PLA reported in the literature (200 g/m^2^ day and 746 cc/m^2^ 24h, respectively) [25] PLA–TEC110 and PLA–TEC–ChNC110 presented significantly better barrier properties. Both WVTR and OTR of PLA–TEC110 were 78% lower than those values reported in literature, owing to its very high crystallinity (49.9%, Table 4). Moreover, with only 1 wt % ChNCs in PLA–TEC–ChNC110, its WVTR and OTR were reduced to 28 g/m^2^.day and 113 cc/m^2^.24h, respectively, which are 36% and 32% lower than those of PLA–TEC110. It is well known that the gas transport properties of polymer composites are greatly affected by a tortuous path and this tortuosity depends on factors such as the shape and aspect ratio of the reinforcement, degree of orientation, and loading of the reinforcement, interface, and crystallinity [28,29,30]. Trifol et al. [31] investigated the effect of nanocellulose and nanoclays on the barrier properties of the PLA. They reported that nanocellulose showed better barrier properties than nanoclays. They attributed this to the increased crystallinity and different shapes of the nanocellulose. In the present study, the improvement of the barrier properties of PLA–TEC–ChNC110 can be attributed to both the good dispersion and distribution of ChNCs and smaller spherulite size that inhibits the permeation of the gas molecules within the polymer matrix. Martinez-Sanz et al. [32] reported that PLA nanocomposites with well-dispersed bacterial cellulose nanowhiskers significantly lowered the water permeability of the PLA nanocomposites.

### 3.5. Hydrolytic Degradation

Hydrolytic degradation tests for PLA110, PLA–TEC110, and PLA–TEC–ChNC110 were performed and the results are presented in Figure 6. As shown in Figure 6a, all three films were degraded and disintegrated after 18 days. With the increase in degradation time, the pH of the aqueous medium of the samples decreased considerably owing to the extraction of lactic acid originating from the PLA. The calculated weight losses of the materials during the tests are presented in Figure 6b. In the initial three days, the degradation of all samples was quite slow, but after one week, the degradation rate increased significantly. Both PLA–TEC110 and PLA–TEC–ChNC110 demonstrated considerably higher degradation rates compared to PLA110, owing to the presence of the TEC plasticizer that increases the free volume of the PLA. It is interesting to note that the nanocomposite degraded slightly faster than PLA–TEC110. The possible reason for this is that the smaller crystallites in the nanocomposite were more easily accessed by water molecules compared to the larger ones in the PLA–TEC110. Similar phenomena have been reported by Paul et al. [34]. They studied the effect of different types of montmorillonites (MMTs) on the hydrolytic degradation of PLA and observed that MMT accelerated the degradation of PLA. The authors concluded that both the composite structure and relative hydrophilicity played vital roles in the hydrolytic degradation of PLA.

The degradation of the polymers in aqueous media proceeds through water uptake followed by chain scission of the ester bond [35]. During degradation, first, the high-molecular-weight PLA chains break down into lower molecular weight chains by the cleavage of the ester bonds of the polymer followed by further disintegration into lactic acid and finally into water and carbon dioxide (see in Figure 7) [36]. Chain scission of the ester bonds is controlled by different parameters such as the amount of water absorbed, the diffusion coefficient of the polymer chain fragment within the polymer, and solubility of the degradation product [37]. The rate of hydrolysis depends on the molecular weight of the oligomers, environmental factors (e.g., temperature), pH of the medium, hydrophilicity, and crystallinity of the given polymer [37,38]. Generally, the hydrolytic degradation of PLA can take place via two different mechanisms: (i) acid hydrolysis (ii) base hydrolysis [39]. Here, hydrolytic degradation is understood to have occurred through acid hydrolysis because the pH substantially decreased from 7 to 3. Acid hydrolysis reactions follow fast chain end sessions and occur by nucleophilic substitution (SN_2_) reactions. The schematic for the acid hydrolyzed degradation of PLA is presented in Figure 7.

#### Effect of Hydrolytic Degradation on Thermal Properties

Hydrolytic degradation affects the thermal properties of PLA110, PLA–TEC110, and PLA–TEC–ChNC110. DSC and TGA curves were recorded before and after hydrolytic degradation and the data are provided in Appendix A (see Appendix A). DSC results show an increase in the degree of crystallinity of the materials after hydrolytic degradation, following the order, PLA110 (61%), PLA–TEC110 (69%), and PLA–TEC–ChNC110 (64%). This was desirable to obtain higher crystallinity after the hydrolytic degradation.

It was observed from the TGA curves that before hydrolytic degradation materials were very stable up to 285 °C. However, after hydrolytic degradation, water has immensely influenced the rate of degradation due to this the materials began degrading much earlier (at 220 °C). DTG curves of all the materials before degradation showed only one peak, whereas those after degradation showed two peaks. These results indicate that water molecules influence the thermal stability of the materials, further confirming the hydrolytic degradation of materials.

### 3.6. Water Uptake Study

In order to investigate the diffusion of water molecules into PLA110, PLA–TEC110, and PLA–TEC–ChNC110 films, the water uptake was recorded at two temperatures; room temperature and at 58 °C. Plots of water uptake vs. time are shown in Figure 8. The water uptake of all the materials was rapid, and the materials were saturated within 1 h. This faster saturation of the materials may be caused by the high crystallinity of the isothermally crystallized films. Generally, an increase in the crystalline domains decreases permeation because high crystallinity reduces the chain mobility and free volume that eventually hinders the attack of water molecules [40]. The water uptake of PLA–TEC–ChNC110 was lower compared to that of PLA110 and PLA–TEC110. This can be attributed to the smaller spherulite size of the ChNC and strong filler-matrix interfacial interaction that may have restricted the water molecules.

The effect of crystallinity on diffusion kinetics of PLA was studied and the data was accumulated in the Table 5. Diffusion behavior is differentiated into three categories *viz.* Case I known as Fickian diffusion, Case II also called as Super Case II, and Case III which is Non-Fickian or anomalous diffusion [41]. The addition of TEC decreases the diffusion coefficient, however, the addition of ChNCs increase the diffusion coefficient. This differentiation is based on the diffusional exponent (n) value. If the value of *n* is 0.5, it will be Fickian diffusion. For Super Case II, *n* > 1 and for Non-Fickian n value varies as 1/2 < *n* < 1.

Table 5 shows that the diffusion exponent for PLA110 and PLA–TEC110 is lower than 0.5, whereas it is greater than 1 for the nanocomposites. This implies that PLA110 and PLA–TEC110 follow the Fickian model, whereas the nanocomposites follow the Super Case II. Higher diffusion exponent (n) value in PLA–TEC–ChNC110 was obviously because of the presence of ChNCs in the nanocomposites. The addition of ChNCs decreases both the water uptake and diffusion coefficient as the spherulites of nanocomposites must have hindered the entry of the water molecules into the nanocomposite films. The mechanism of entry of water molecules into the nanocomposite film is illustrated in Figure 9. Chow et al. [41], studied the effect of organo-montmorillonite (OMMT) and nano-precipitated calcium carbonate (NPCC) on the water absorption of PLAs, and found that the n values for PLA, OMMT, and NPCC were in the range of 0.25–0.38 i.e., below 0.5, indicating that the Fickian diffusion model (i.e., Case I) was followed.

## 4. Conclusions

In this study, neat PLA, plasticized PLA–TEC, and plasticized PLA nanocomposites with 1 wt % of ChNC (PLA–TEC–ChNC) were prepared with liquid-assisted extrusion. Compression molding was performed at varying temperatures and holding times to prepare films with different crystallinities. The effect of well dispersed and distributed ChNC on the crystallinity and properties of PLA nanocomposites was investigated.

Dispersion of ChNC into PLA-TEC was examined by SEM and it was observed that nanocomposites exhibited homogenous dispersion and distribution of ChNC. It is noticeable that PLA–TEC–ChNC110 exhibited very high crystallinity (48%) within 5 min of crystallization as confirmed by DSC which is ascribed due to the good dispersion of the ChNC. On the other hand, PLA110 and PLA–TEC110 required 40 min to achieve similar crystallinity of 45% and 49%, respectively.

Addition of properly dispersed ChNCs increased the nucleation ability which further affected the crystallite size of PLA as examined by XRD analysis. The crystallite size of nanocomposites was smaller (8 nm) than that of the neat PLA (12 nm) and PLA–TEC (14 nm). In addition, the spherulites sizes of the PLA110, PLA–TEC110, and PLA–TEC–ChNC110 were determined by POM as 49, 56, and 23 µm, respectively. The spherulites size affected the optical properties of the films; consequently, the light transmittance of PLA110, PLA–TEC110, and PLA–TEC–ChNC110 was determined to be 61%, 58% and 69%, respectively. Owing to smaller spherulites of PLA–TEC–ChNC110, light can pass through the films; therefore, the transmittance of nanocomposites is higher than PLA110 and PLA–TEC110.

It was found that ChNCs significantly reduced the water and oxygen barrier properties of PLA–TEC110. WVTR and OTR of PLA–TEC110 were, respectively, 36% and 32%, lower in comparison to PLA–TEC–ChNC110. The positive effect of ChNCs on the barrier properties is attributed to be due to the good dispersion, better nucleation ability and smaller spherulite size. Crystallinity caused a decrease in water diffusion. Increased crystallinity and ChNC strongly affected the hydrolytic degradation of the PLA.

This study provides a prominent enhancement in the properties of plasticized PLA when 1 wt % ChNCs are homogenously mixed in PLA–TEC. The knowledge gained from this study is expected to be helpful for the preparation of materials for packaging applications.

## Figures and Tables

**Figure 1 polymers-12-00726-f001:**
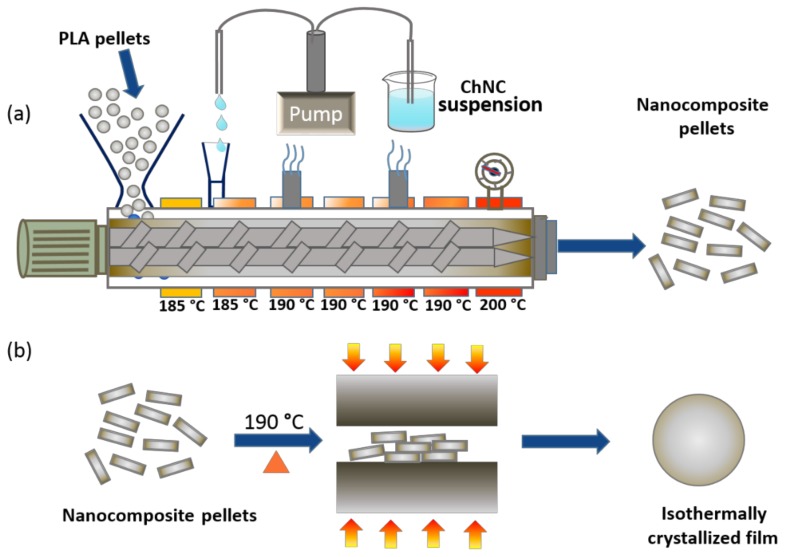
Schematics of (**a**) preparation of nanocomposites pellets (PLA–TEC–ChNC) via liquid-assisted extrusion process, and (**b**) preparation of isothermally crystallized nanocomposite films.

**Figure 2 polymers-12-00726-f002:**
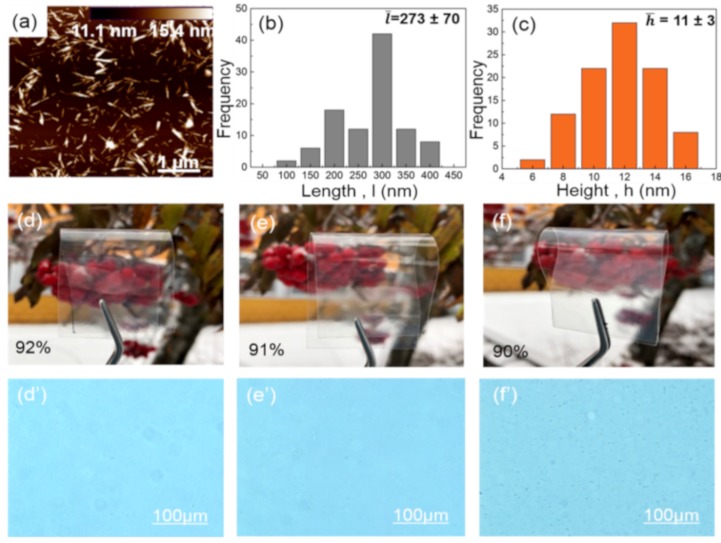
ChNCs characteristics including (**a**) height AFM image showing shape of ChNCs, (**b**) length and (**c**) diameter (height) distributions indicating average length (Ī) and width (ħ). Photographs and optical micrographs of neat (**d**–**d’**) PLA, (**e**–**e’**) PLA–TEC, and (**f**–**f’**) PLA–TEC–ChNC films showing high transparency and well-dispersed ChNCs in PLA–TEC.

**Figure 3 polymers-12-00726-f003:**
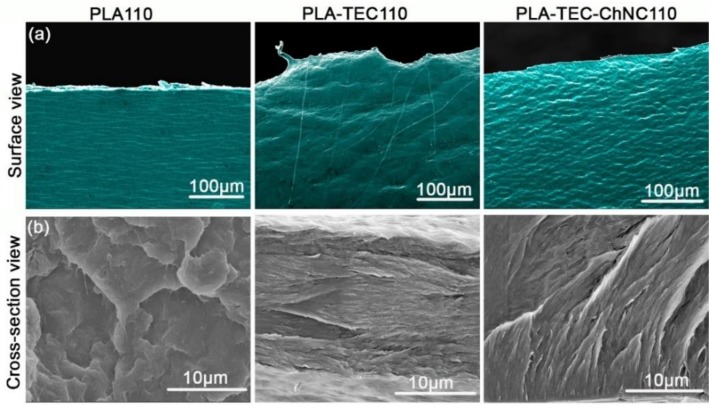
SEM images of fractured samples of isothermally crystallized PLA110, PLA–TEC110, and PLA–TEC–ChNC110 films (**a**) surface view and (**b**) cross-sectional view.

**Figure 4 polymers-12-00726-f004:**
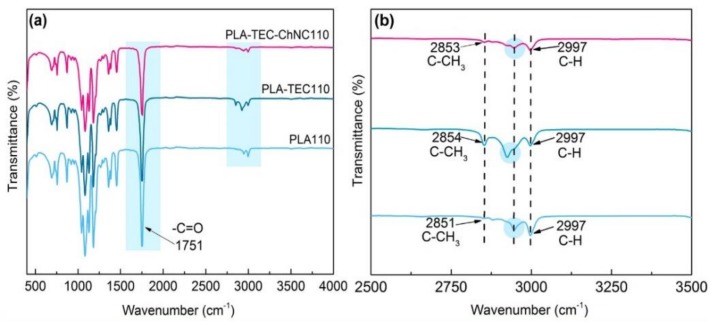
ATR-FT-IR spectra of (**a**) PLA110, PLA–TEC110, and PLA–TEC–ChNC110 (**b**) zoomed view of PLA, PLA–TEC110, and PLA–TEC–ChNC110 showing the –C–CH_3_ and –C–H peaks.

**Figure 5 polymers-12-00726-f005:**
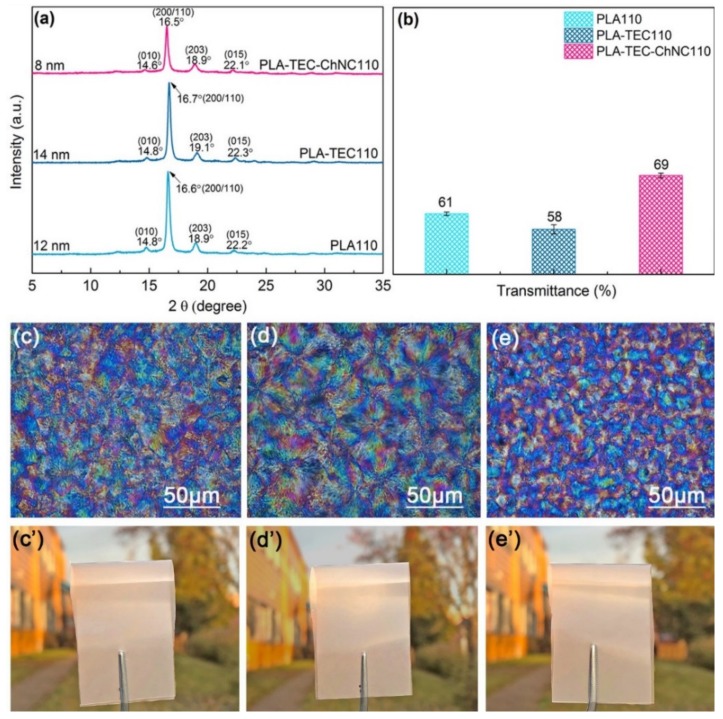
(**a**) XRD and (**b**) UV spectra at 550 nm, optical micrographs and photographs of isothermally crystallized (**c**–**c’**) PLA110, (**d**–**d’**) PLA–TEC110, and (**e**–**e’**) PLA–TEC–ChNC110.

**Figure 6 polymers-12-00726-f006:**
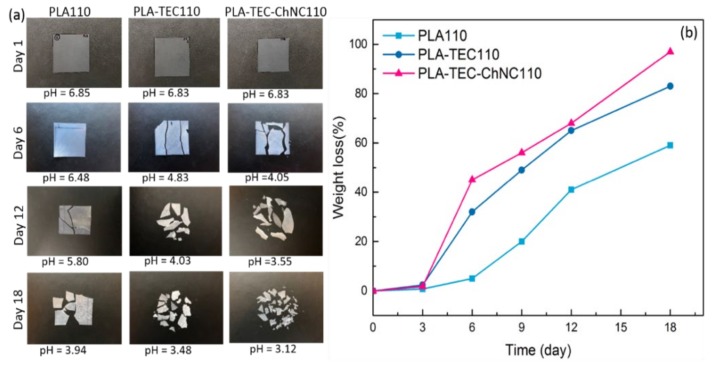
Visual changes in PLA110, PLA–TEC110, and PLA–TEC–ChNC110 after 18 days of hydrolytic degradation (performed at 58 °C). (**a**) Degradation greatly affected the pH and (**b**) chart showing the degradation of PLA110, PLA–TEC110, and PLA–TEC–ChNC110.

**Figure 7 polymers-12-00726-f007:**
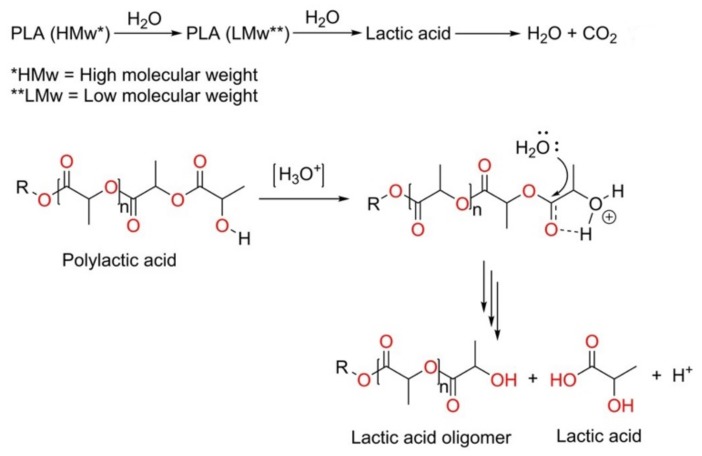
Mechanism of hydrolysis of PLA showing how chain secession of PLA occurs in the acidic medium; here, R=CH_3_.

**Figure 8 polymers-12-00726-f008:**
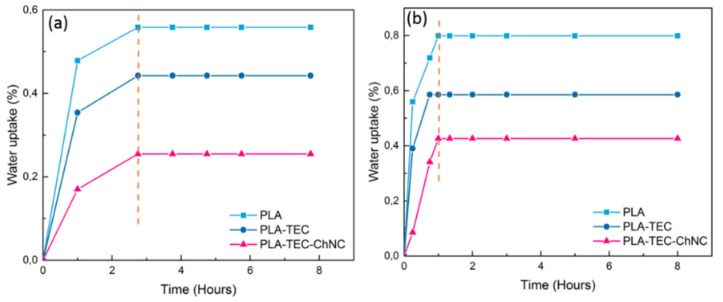
Water uptake as a function of time showing the effect of isothermal crystallization on PLA, PLA–TEC, and PLA–TEC–ChNC films (**a**) at room temperature and (**b**) at 58 °C.

**Figure 9 polymers-12-00726-f009:**
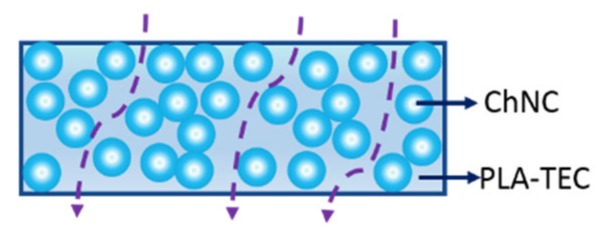
Schematics showing how water enters in PLA–TEC–ChNC110; spherulites developed in the nanocomposites are hindering the water molecules and forming a tortuous path.

**Table 1 polymers-12-00726-t001:** Sample codes and compositions of prepared materials.

Materials	Feeding Rate (kg/h)		Composition of Materials (wt %)		
	PLA	Suspension	PLA	TEC	ChNCs
PLA	2.00	0.00	100	0	0
PLA-TEC	1.80	0.75 *	90	10	0
PLA-TEC-ChNC	1.78	0.77 *	89	10	1

* Fed into extruder out of which 0.09 kg/h of water and 0.46 kg/h ethanol were removed as vapor during extrusion.

**Table 2 polymers-12-00726-t002:** Temperature and time used for preparation of different isothermally crystallized films of PLA, PLA–TEC, and PLA–TEC–ChNC.

Sample Codes	*IC*_TEMP_ (°C)	*IC*_TIME_ (min)
PLA	N/A	0
PLA–TEC	N/A	0
PLA–TEC–ChNC	N/A	0
PLA135–5	135	5 *
PLA–TEC135–5	135	5 *
PLA–TEC–ChNC135–5	135	5 *
PLA135–15	135	15 *
PLA–TEC135–15	135	15 *
PLA–TEC–ChNC135–15	135	15 *
PLA130–5	130	5 *
PLA–TEC130–5	130	5 *
PLA–TEC–ChNC130–5	130	5 *
PLA130–15	130	15 *
PLA–TEC130–15	130	15 *
PLA–TEC–ChNC130–15	130	15 *
PLA125	125	5 *
PLA–TEC125	125	5 *
PLA–TEC–ChNC125	125	5 *
PLA125	125	15 *
PLA–TEC125	125	15 *
PLA–TEC–ChNC125	125	15 *
PLA110	110	40 **
PLA–TEC110	110	40 **
PLA–TEC–ChNC110	110	5 **

Note: * At 5 and 15 min, all the materials exhibited incomplete crystallization. ** For further experiments, complete and homogenous crystallization has been considered (which was achieved at 110 °C), PLA110 and PLA–TEC110 took 40 min while PLA–TEC–ChNC110 was completely crystallized within 5 min.

**Table 3 polymers-12-00726-t003:** Thermal properties of neat PLA, PLA–TEC, and PLA–TEC–ChNC, and isothermally crystallized PLA110, PLA–TEC110, and PLA–TEC–ChNC110 films.

Materials	*T*_g_ (°C)	*T*_cc_ (°C)	*T*_m_ (°C)	Crystallinity (%)
PLA	61.7	110.4	170.4	4.0
PLA–TEC	48.8	98.0	164.8	6.6
PLA–TEC–ChNC	48.1	95.4	164.4	6.7
PLA135-5	58.9	108.6	169.6	7.5
PLA–TEC135–5	46.0	97.4	164.6	7.8
PLA–TEC–ChNC135–5	46.3	94.7	163.2	8.6
PLA135–15	60.6	110.6	170.0	1.3
PLA–TEC135–15	47.5	97.4	164.4	4.5
PLA–TEC–ChNC135–15	44.8	93.4	163.4	14.2
PLA130–5	59.4	108.4	169.4	2.6
PLA–TEC130–5	46.8	96.2	164.5	7.5
PLA–TEC–ChNC130–5	46.3	94.5	163.6	8.2
PLA130–15	58.6	108.5	168.6	7.0
PLA–TEC130–15	45.7	96.0	163.3	7.2
PLA–TEC–ChNC130–15	39.1	87.1	163.3	34.5
PLA125–5	59.5	109.7	169.5	6.0
PLA–TEC125–5	46.9	97.0	164.8	7.2
PLA–TEC–ChNC125–5	46.5	95.8	164.7	7.3
PLA125–15	59.8	110.4	169.6	5.8
PLA–TEC125–15	47.3	96.9	163.7	6.0
PLA–TEC–ChNC125–15	37.7	86.9	162.7	36.5
PLA110–40	59.4	111.8	164.9	45.5
PLA–TEC110–40	44.7	113.7	164.8	49.9
PLA–TEC–ChNC110–5	36.8	-	164.6	47.5

**Table 4 polymers-12-00726-t004:** Comparison of WVTR, OTR, and OP of PLA, PLA110, and PLA–TEC–ChNC110. (PLA data is from references [25,33])

Materials	WVTR [g/m^2^ day]		OTR [ml/(m^2^ 24h)]		OP [(ml μm)/(m^2^ 24h kPa)]	
	Ave	std	Ave	std	Ave	std
PLA	200 [25]	-	746 [33]	-	-	-
PLA–TEC110	44	(3)	165	(21)	19,425	(3076)
PLA–TEC–ChNC110	28	(2)	113	(18)	11,563	(2210)

**Table 5 polymers-12-00726-t005:** Kinetics of water uptake of the PLA110, PLA–TEC110, and PLA–TEC–ChNC110.

Materials	Diffusional Exponent (n)	Kinetic Constant (K)	Diffusion Coefficient (D) (m^2^ s^−1^)
PLA110	0.248	0.909	1.62 × 10^−6^
PLA–TEC110	0.315	1.095	1.09 × 10^−6^
PLA–TEC–ChNC110	1.230	0.979	0.10 × 10^−6^

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
