# Peer review of "Effect of Chitin Nanocrystals on Crystallization and Properties of Poly(lactic acid)-Based Nanocomposites"

_polymers, 2020, doi:10.3390/polym12030726_

Round 1

Reviewer 1 Report

The manuscript reports an scalable method for preparation of PLLA/nanocellulose composites and studied the crystallization, barrier and degradation properties of the produced films. The results show that well dispersion of nanocellulose can enhance the nucleation rate, barrier properties and hydrolytic degradation rate.

The work deserves publication in Polymer after considering the following points:

The preparation of PLLA nanocomposites via liquid (containing water) assisted extrusion process might lead to serious hydrolytic degradation of PLA during the extrusion process. The authors need check the Mw of PLLA after extrusion.

The last sentences on page 8: “PLA–TEC–ChNC110 did not show significant differences but the overall intensity of the nanocomposites decreased and some overlapping bands were found in the fingerprint region. This may be due to certain molecular interactions that may have occurred between the components of the nanocomposites.” Since there is no apparent shift of absorbance band (e. g. C=O band), there are not considerable intermolecular interactions and the decreased intensity might result from the different film thickness.

Some data in Table 3 is strange. For example, PLA135-15 sample shows a lower degree of crystallinity than PLA135-5. It is abnormal that the sample isothermally crystallized for a longer time has a lower crystallinity than that for a short time. In addition, there are two “PLA135-5”. The second should be “PLA130-5”.

Table 3 shows that ChNCs demonstrates an excellent nucleation ability for PLLA. What is the reason?

Line 322 on page 10: “Spherulites sizes of the PLA110, PLA–TEC110, and PLA–323 TEC–ChNC110 were measured to be 49 μm, 56 μm, and 23 μm, respectively, which are consistent with the XRD results.” The assertion is not reasonable since WAXD cannot distinguish spherulites with size of microns.

In Table 4, PLA-TEC-ChNC110 shows an exponent of 1.230, namely super diffusion case. This is quite different from the other two counterparts and the previous reports on nanocomposites. The authors should comment on this interesting result.

Author Response

Please see the separate response file.

Reviewer 2 Report

Review of the manuscript “Effect of chitin nanocrystals on crystallization and properties of poly(lactic acid)-based nanocomposites” by Singh, Patel, Schwendemann, Zaccone, Geng, Maspoch and Oksman

In this paper nanocomposites of plasticized PLA and chitin nanocrystals (1 wt%) were prepared via liquid-assisted extrusion. Nanocomposite films with different crystallinities were obtained by compression molding at different temperatures and holding times, showing good dispersion of nanocrystals, as assessed by SEM. Subsequently, thermal optical, barrier and hydrolytic degradation properties were measured. For nanocomposites a nucleation effect of chitin nanocrystals was measured by DSC and an smaller spherulite size, with comparison with PLA and plasticized PLA, was obtained. As consequence, a reduction in water vapor transmission rate and oxygen permeability were measured. These results, together with the increase in hydrolytic degradation rate showed by nanocomposites prove that they can be considered to be used for packaging applications.

In my opinion, the manuscript is within the scope of the journal and is worthy of publication. As consequence, I would recommend it for publication after minor amendment that I explain in the following:

1- Section 3.5.1. I think that more detailed description of TGA curves of PLA110, PLA-TEC110 and PLA-TEC-ChNC110 before degradation should be included.

2- Table 4. Units of diffusion coefficient should be indicated.

3- Supplementary Material: Figure S1 c) and d). I would recommend using dotted curves for DTG curves in order to make easier to distinguish between TGA and DTG curves.

Author Response

see a attached response file
